# Analysis of the Effects of Nutrient Intake and Dietary Habits on Depression in Korean Adults

**DOI:** 10.3390/nu13041360

**Published:** 2021-04-19

**Authors:** Hyeonseo Yun, Dong-Wook Kim, Eun-Joo Lee, Jinmyung Jung, Sunyong Yoo

**Affiliations:** 1School of Electronics and Computer Engineering, Chonnam National University, Gwangju 61186, Korea; 0525yhs@gmail.com; 2Big Data Steering Department, National Health Insurance Service, Wonju 26464, Korea; kimdw@nhis.or.kr (D.-W.K.); eejj1543@naver.com (E.-J.L.); 3Division of Data Science, College of Information and Communication Technology, The University of Suwon, Hwaseong 18323, Korea; 4Department of ICT Convergence System Engineering, Chonnam National University, Gwangju 61186, Korea

**Keywords:** depression, nutrient intake, dietary habits, National Health and Nutrition Examination Survey

## Abstract

While several studies have explored nutrient intake and dietary habits associated with depression, few studies have reflected recent trends and demographic factors. Therefore, we examined how nutrient intake and eating habits are associated with depression, according to gender and age. We performed simple and multiple regressions using nationally representative samples of 10,106 subjects from the Korea National Health and Nutrition Examination Survey. The results indicated that cholesterol, dietary fiber, sodium, frequency of breakfast, lunch, dinner, and eating out were significantly associated with depression (*p*-value < 0.05). Moreover, depression was associated with nutrient intake and dietary habits by gender and age group: sugar, breakfast, lunch, and eating out frequency in the young women’s group; sodium and lunch frequency among middle-age men; dietary fibers, breakfast, and eating out frequency among middle-age women; energy, moisture, carbohydrate, lunch, and dinner frequency in late middle-age men; breakfast and lunch frequency among late middle-age women; vitamin A, carotene, lunch, and eating out frequency among older age men; and fat, saturated fatty acids, omega-3 fatty acid, omega-6 fatty acid, and eating out frequency among the older age women’s group (*p*-value < 0.05). This study can be used to establish dietary strategies for depression prevention, considering gender and age.

## 1. Introduction

Depression is a common and serious mental disorder that affects over 264 million people of all ages worldwide [1]. It continues to attract attention as a cause of many emotional and physical problems. According to the U.S. Census Bureau’s Household Pulse Survey for 2020, 34% of respondents living in the U.S. had symptoms of depression and anxiety [2]. According to the survey by the Ministry of Health and Welfare in Korea, the number of depressed patients of all age groups was 680,169 in 2017, 751,930 in 2018, and 798,000 in 2019. These numbers are increasing every year, which is a serious social problem. In addition, as depression is a direct and indirect cause of other diseases and suicide, it requires timely treatment [3,4,5,6,7]. However, treatment is difficult for complex reasons, such as negative social evaluation and a high recurrence rate [8,9,10,11,12]. Therefore, it is necessary to identify various factors related to depression, and establish measures to prevent and manage depression in advance.

Various economic and social factors, such as rapid economic growth in Korean society and the increase of single-person households, have led to changes in dietary habits [13]. The growth of multicultural foods due to the globalization of diets, and increased processed food intake, have changed nutrient intake overall [14,15,16]. The culture of dining twice a day or alone has also changed nutrient intake and dietary patterns [17,18]. Interestingly, previous studies have demonstrated that changes in nutrient intake and dietary patterns affect mental health [16,19,20,21]. Epidemiological studies have shown that a low intake of nutrients such as protein, dietary fiber, potassium, and vitamin C, along with breakfast fasting, are associated with depression [22,23,24,25]. Observational and experimental studies have shown that food intake, vitamin B, energy, and omega-3 fatty acid intake are associated with depression [26,27,28]. In fact, depression can be effectively prevented and alleviated by improving dietary habits [29,30]. Therefore, it is necessary to analyze the characteristics of nutrient intake and dietary habits in order to prevent depression. However, most previous studies do not reflect recent trends of nutrient intake and dietary habits. Considering that eating habits acquire different attributes each year, our analysis uses recent data from 2014 to 2018 [13]. Furthermore, existing studies have analyzed specific age or gender groups. Follow-up research considering all ages and genders is necessary to address this gap. Finally, previous studies have conducted analyses focused on certain nutrients. However, the amount and type of nutrients are often different in real life, and the interaction between nutrients should be considered [31,32]. In this case, a comprehensive analysis using various nutritional variables is required.

This study aims to analyze the effects of nutrient intake and eating habits according to demographic factors on the severity of depression. Based on data from the 6th (2014) and 7th (2016, 2018) Korea National Health and Nutrition Examination Surveys (K-NHANES), statistical analysis was conducted to identify factors associated with depression [33,34,35]. Subjects were grouped according to demographic factors such as gender and age group. The novelty of our study is threefold: (1) we reflect on relatively recent dietary trends; (2) we analyze the association between nutrient intake and depression by gender and age, based on the severity of depression, not on dichotomous approaches; and (3) we perform a comprehensive analysis using various nutrients and dietary variables.

## 2. Materials and Methods

### 2.1. Study Population

We utilized secondary data from the 6th (2014) and 7th (2016, 2018) K-NHANES of the Korea Centre for Disease Control and Prevention (KCDC). The K-NHANES is conducted to evaluate the status of people’s health and nutritional conditions, and consists of a health examination, health interview, and dietary data. Use of the K-NHANES was approved by the institutional review board of the KCDC (2013-12EXP-03-5C, 2018-01-03-P-A).

The K-NHANES is a nationally representative survey, which uses two-stage stratified cluster sampling for a complex sampling design [36,37]. To account for the complex sampling design, we used strata, clusters, and weights. Such variables were applied to enhance the representative nature of the sample. We calculated the combined weights for the 6th and 7th K-NHANES. Next, we integrated the 6th and 7th K-NHANES data based on the combined weights to maintain sample representation. The procedures for handling the data are shown in Figure 1. First, 19,849 adults aged 20 years or older were selected from the K-NHANES. We excluded participants with incomplete dietary data or depression questionnaires. Finally, a total of 10,106 participants were selected for our study.

### 2.2. Study Variables

We used the depression questionnaire as a dependent variable, which was constructed based on the Patient Health Questionnaire-9 (PHQ-9). The PHQ-9 consists of nine questions corresponding to the diagnostic criteria for major depression episodes [38]. This test has the advantage of being able to monitor the severity and response to treatment of major depressive disorder in a relatively short time period. In the K-NHANES, the PHQ-9 questionnaire was standardized for Korean society. The PHQ-9 scores range from 0 to 27, and higher scores indicate a higher risk of depression. Therefore, we used PHQ-9 scores to represent the severity of depression. Nutrient intake and dietary habit information of the K-NHANES were used as independent variables. From the nutrient intake information, we extracted 23 variables: energy, moisture, protein, fat, saturated fatty acids, monounsaturated fatty acids, omega-3 fatty acids, omega-6 fatty acids, cholesterol, carbohydrates, dietary fiber, sugars, calcium, phosphorus, iron, sodium, potassium, vitamin A (retinol equivalent), carotene, vitamin B1 (thiamine), vitamin B2 (riboflavin), vitamin B3 (niacin), and vitamin C intake. From the dietary habit information, we extracted four variables, including breakfast frequency in the past year, lunch frequency in the past year, dinner frequency in the past year, and the number of meals eaten out in the past year. The demographic variables, including gender and age, were used for the classification of subgroups. Subjects were divided by gender into the following age groups: 19–34 years old, 35–49 years old, 50–64 years old, and 65 years old or older.

### 2.3. Statistical Analysis

Statistical analysis, including means, standard deviation, and 95% confidence intervals, were used to describe the basic characteristics of gender groups. A *t*-test was performed to confirm statistically significant mean distributions of gender groups [39]. Simple linear regression was used to identify statistically significant variables of PHQ-9 scores and eating habits. However, simple linear regression causes bias in regression coefficients by omitting important independent variables [40,41]. To overcome this limitation, we used univariate multiple linear regression and examined the regression coefficient, *p*-value, and 95% confidence interval (CI) [42]. In this step, we used the variables identified in simple linear regression as independent variables. Additionally, we performed multiple linear regression in subgroups to identify significant variables according to gender and age. For the categorical variables, we used the group with the largest number of samples as the reference group. All statistical analyses were performed using SAS version 9.4 (SAS Institute, Cary, NC, USA). The SURVEYMEAN procedure was conducted to examine descriptive statistics related to eating habits and the severity of depressive symptoms. The SURVEYREG procedure was conducted to examine the *t*-test, simple linear regression, and multiple linear regression for eating habits and PHQ-9 scores. The statistical significance testing was based on a *p*-value < 0.05. Furthermore, we performed multiple linear regressions on the three reduced factors that were derived from the 23 nutrient intake variables by factor analysis. The objective of the factor analysis was to convert sets of correlated variables into a few non-correlated variables, which allows for more accurate models [43,44]. The multiple linear regression models along with the three factors were applied to the men, women, and overall groups.

## 3. Results

### 3.1. Basic Characteristics of the Study Population

The basic characteristics of the study population gender groups are shown in Table 1. Both the men’s and women’s groups had a predominance of ages 35–49 (15.3% and 14.6%), were married (71.0% and 81.7%), had a high-income (25.8% and 25.5%), and had an upper college education (46.2% and 37.4%). There were differences in the gender groups: the women’s group had a lower food intake (mean = 1391.6 g) than the men’s group (mean = 1834.9 g), and lower frequency of lunch and dinner (5–7/week, 87.6% and 86.9%) than the men’s group (5–7/week, 91.5% and 92.9%). Further, the women’s group had a lower frequency of eating out (2>/day, 3.8%) than the men’s group (2>/day, 14.0%). The men’s group had a lower frequency of breakfast (5–7/week, 58.9%) than the women’s group (5–7/week, 60.6%), while the women’s group had higher frequency of skipping lunch and dinner (<5/week, 1.6% and 0.2%) than the men’s group (<5/week, 1.8% and 0.5%). Men had a higher frequency of skipping breakfast (<1/week, 16.4%) than women (<1/week, 15%). All nutrient intakes were statistically significant (*p*-value < 0.05) in the *t*-test between gender groups. Frequency of lunch, dinner, and eating out were statistically significant (*p*-value < 0.05). The basic characteristics of the study population demonstrate two things. First, dietary habits show differences based on gender. Second, most nutrient intake and dietary habits were found to be statistically significantly different between gender groups.

The age-related changes of depression severity in men and women are shown in Figure 2. In the men’ group, depression severity reached its highest value in the age range of 20–34, and decreased with increasing age, then increased again at 65 years or older. Similarly, in the women’ group, depression severity reached its highest value at the age range of 20–34, decreased with increasing age, then increased again at 50 years or older.

### 3.2. Nutritional Intake and Eating Habits of the Study Population

The univariate simple linear regression results of nutrient intake and dietary habits as they relate to PHQ-9 scores are shown in Table 2. The results show that all nutrient intakes except iron were statistically significant. We found that energy, moisture, protein, fat, saturated fatty acids, monounsaturated fatty acids, polyunsaturated fatty acid, omega-3 fatty acids, omega-6 fatty acids, cholesterol, carbohydrates, dietary fiber, calcium, phosphorus, sodium, potassium, vitamin A, carotene, vitamin B1, vitamin B2, vitamin B3, and vitamin C were statistically significant (*p*-value < 0.05). Among dietary habits, the overall frequency of breakfast, lunch, dinner and eating out were statistically significant, while that of lunch (3–4/week), dinner (0/week), and eating out (3–4/week) frequency tended to be less significant (*p*-value ≥ 0.05). Next, we performed a univariate multiple regression. Unnecessary variables in the multiple linear regression adversely affect the performance of the model [42]. This problem can be solved by using statistically significant variables in simple linear regression [45]. Therefore, we performed the multiple linear regression using statistically significant variables from the simple linear regression. The results revealed that cholesterol, dietary fiber, sodium, frequency of breakfast (1–2/week, 0/week), lunch (3–4/week, 1–2/week), dinner (1–2/week), and eating out (1</month) were statistically significant (*p*-value < 0.05).

### 3.3. Further Analysis Based on the Demorgraphic Characteristics

The multiple linear regressions for subgroups of gender and age are shown in Table 3 and Table 4. The young women’s group showed an association with sugar, breakfast (1–2/week, 0/week), lunch (1–2/week), and eating out (1</month) frequency (*p*-value < 0.05). The young men’s group did not show any association with nutrient intake and dietary habits. The middle-age groups had different associations in nutrient intake and dietary habits. The middle-age men’s group showed that sodium and lunch frequency (3–4/week) were statistically significant (*p*-value < 0.05). In the middle-age women’s group, dietary fiber, breakfast (0/week), and eating out (1/day, 5–6/week) frequency were statistically significant (*p*-value < 0.05). In the late middle-age men’s group, energy, moisture, carbohydrates, lunch (3–4/week), and dinner (3–4/week, 1–2/week) frequency were statistically significant (*p*-value < 0.05). In the late middle-age women’s group, breakfast (0/week) and lunch (3–4/week, 1–2/week) frequency were statistically significant (*p*-value < 0.05). The older age men’s group showed statistical significance in vitamin A, carotene, lunch (1–2/week), and eating out (1</month) frequency (*p*-value < 0.05), while in the older age women’s group, fat, saturated fatty acids, omega-3 fatty acid, omega-6 fatty acid, and frequency of eating out (2>/day) were statistically significant (*p*-value < 0.05).

The coefficients in the multiple regression models across the eight subgroups were also described in Appendix A as the heatmap with their significance. We observed that the significant variables in two or more subgroups were eating out (1</month), lunch (1–2/week), lunch (3–4/week), and breakfast (0/week) frequency. Most variables were not commonly determined as significant among the subgroups, which indicates that different variables should be considered with depression for each sub group.

### 3.4. Multiple Linear Regression with Factor Analysis on Nutrient Intake Variables

The analyzed factors based on the 23 nutrient variables and their factor scores, are described in Table 5. Twelve nutrient variables (fat, monounsaturated fatty acids, saturated fatty acids, protein, omega-6 fatty acids, cholesterol, energy, vitamin B2, phosphorus, vitamin B3, vitamin B1, and omega-3 fatty acids) presented the highest scores at factor1. Nine nutrient variables (dietary fiber, carbohydrates, potassium, moisture, sugar, iron, sodium, calcium, and vitamin C) showed the most significant scores at factor2. In addition, two variables (vitamin A and carotene) had the highest scores at factor3.

When multiple linear regression was applied to the entire, men, and women groups along with the three factors, significant coefficients were observed at factor2 and 3 in the entire and women’s groups. In the men’s group, only factor3 was significant (Table 6).

## 4. Discussion

This study provides evidence that depression is associated with nutrient intakes and eating habits. Sugar, sodium, vitamin A, carotene, moisture, fat, saturated fatty acids, omega-3 fatty acid, omega-6 fatty acid, dietary fiber, and frequency of breakfast, lunch, dinner, and eating out were significantly associated with depression. Depression was found to be associated with different dietary habits based on gender and age.

Previous studies have revealed an association between depression and food intake, specifically the intake of protein, dietary fiber, potassium, vitamin B, vitamin C, omega-3 fatty acids, and saturated fatty acids, along with skipping breakfast [22,23,24,25,26,27,28,46,47]. In line with previous studies, our study demonstrated that depression is associated with the following: dietary fiber for all participants; breakfast frequency in the young, middle-age women’s and late middle-age women’s group; and saturated fatty acids, and omega-3 fatty acids in older age women’s group. Contrary to previous findings, we did not find any relationship with protein, potassium, vitamin B, and vitamin C intakes in any group. Interestingly, we found that sodium, cholesterol, and frequency of lunch, dinner, and eating out were associated with depression. We also observed the following additional relationships: sugar, lunch, and eating out frequency in the young women’s group; sodium and lunch frequency in the middle-age men’s group; eating out frequency in the middle-age women’s group; energy, moisture, carbohydrates, dinner frequency in the late middle-age men’s groups; eating out frequency in older groups; vitamin A, carotene, and lunch frequency in the older men’s group; and fat and omega-6 fatty acid in the older women’s group. These results suggest that the relationship between eating habits and the severity of depression varies according to gender and age.

The factor1 in the factor analysis could be named as ‘Fat and vitamin B’ group as all fat and vitamin related variables are highly associated to the factor1. This association could be supported by Baltaci’s work that reports the correlation between fat and vitamin B [48]. The cholesterol is also included in the factor1 with high score, which is consistent with the fact that dietary fats intake and cholesterol are strongly correlated [49]. The factor3 is associated with vitamin A and carotene, which can be easily expected because carotene is a member of the vitamin A family [50]. The variable vitamin A and carotene are determined as significant in the older age men’s group, which could be of one of reasons for the significance of factor3 in the men’s group. In addition, the variable sugar in the young women’s group and the variable dietary fiber in the late middle-age women’s group are resulted as significant. It can be one explanation for the significance of factor2 in the women’s group.

We found some evidence in the literature of plausible mechanisms linking depression and the significant variables in this study. In Swann’s work, a mechanism between depression and dietary fiber, as one of the significant variable in the multiple regression applied to the entire group, was summarized as follows [51]: Dietary fiber intake → healthy microbiota → increased short-chain fatty acids → inhibition histone deacetylase or G-protein–coupled receptor activation → decreased depression. This negative correlation between dietary fiber intake and depression was also supported by several other studies [52,53]. Furthermore, a possible mechanism of depression associated with sodium, another significant variable in the multiple regression applied to the entire group, was presented in Ozdemir’s work, as follows [54]: Serum intake → thirst → arginine vasopression (AVP) release → adrenocorticotropic hormone (ACTH) release → activation of the hypothalamic–pituitary–adrenal (HPA) function → decreased depression. The negative correlation between depression and serum intake was also addressed in Özdemir’s work [55]. In accordance with these previous studies, the proposed multiple regression model produced negative coefficients for dietary fiber and serum intake.

Our study has limitations that are common to most epidemiological studies. We found that eating habits and nutrition intake are associated with depressive symptoms, but this does not imply causality [56]. Observational and experimental studies or longitudinal studies are necessary to confirm a causal relationship. Another limitation of this study is that we applied only age and gender, which are basic demographic factors. Considering the complexity of the depression mechanism, future research should address this issue by using other demographic factors such as income, educational level, and occupation.

Depression is an important field for research into prevention because it includes various problems—illness and suicide risks, difficulties in diagnosis and treatment, and complex mechanisms of occurrence. Changes in eating habits over time have also been associated with depression. This study is the first to show the relationship between eating habits and the severity of depression in 2014, 2016, and 2018. When comparing our results to those of previous studies, notably, eating habit trends in 2018 are not included. Therefore, this study contributes to a timely discussion of the association between eating habits and the severity of depression. Additionally, through the use of demographic factors for the analysis, these results go beyond previous studies, providing new evidence of eating habits and the severity of depression according to gender and age. This study provides insights into the prevention depression based on eating habit information during the life cycle. It can be used to improve diets with nutrient intake and dietary patterns information according to gender and age. Furthermore, eating habits can predict depression, and be used for early diagnosis and treatment of depression. This study can be combined with other mechanisms of depression occurrence, and may lead to further studies on depression prevention.

## 5. Conclusions

This study suggests there are association between eating habits and the severity of depression. We examined different associations between eating habits and the severity of depression by gender and age. Notably, our results revealed new relationships between eating habits and the severity of depression by gender and age. Considering the changes in eating habits, interventions to improve diet may not only prevent depression, but also reduce the cost of treating depression. Future observational and experimental studies or longitudinal studies should be conducted to verify these associations.

## Figures and Tables

**Figure 1 nutrients-13-01360-f001:**
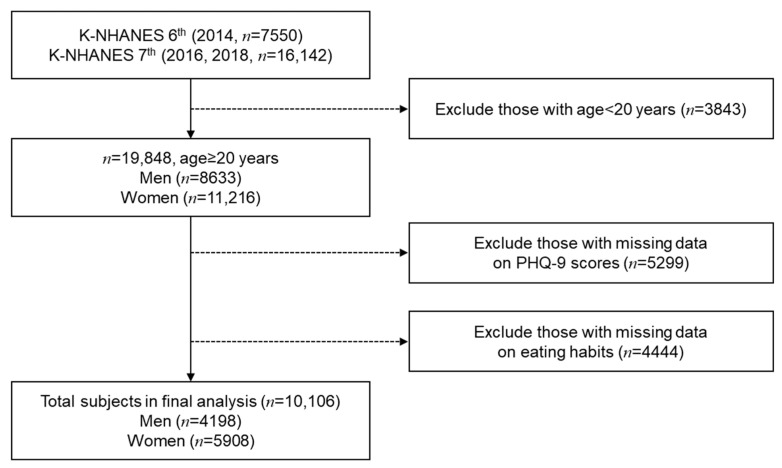
Flow diagram for selecting participants of the study.

**Figure 2 nutrients-13-01360-f002:**
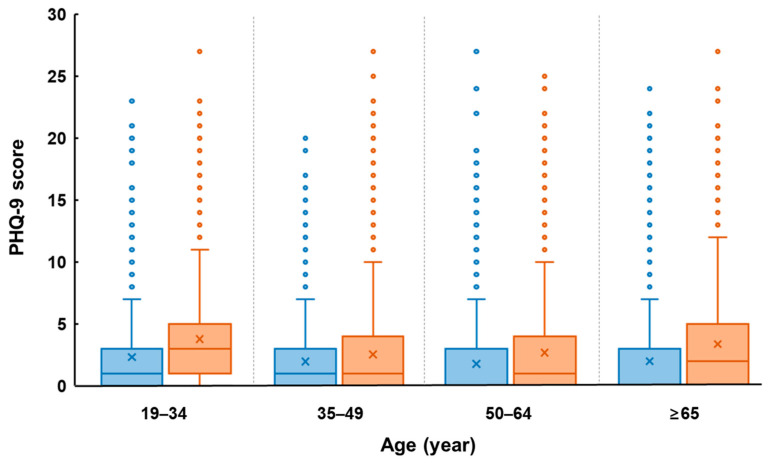
A box and whisker plot of PHQ-9 scores by gender and age. The horizontal line inside the box represents median, and the ‘x’ symbol represents mean.

**Table 1 nutrients-13-01360-t001:** Basic characteristics of the study population.

Variables	Men (*N* = 4198)	Women (*N* = 5908)	*p*-Value
**Demographic factor (*N* (%))**			
Age (categorical)			<0.0001
19–34	809 (13.9)	1057 (12.5)	
35–49	1129 (15.3)	1707 (14.6)	
50–64	1129 (12.7)	1649 (13.6)	
Over 65	1131 (7.8)	1495 (9.6)	
Marital Status			<0.0001
Married	3339 (71.0)	5122 (81.7)	
Single	848 (29.0)	769 (18.3)	
Income level			0.8587
Low	1032 (25.0)	1444 (24.8)	
Low intermediate	1024 (24.3)	1487 (24.8)	
High intermediate	1057 (24.9)	1473 (24.9)	
High	1074 (25.8)	1487 (25.5)	
Education level			<0.0001
Below elementary school	623 (9.5)	1460 (19)	
Middle school	421 (7.5)	588 (8.9)	
High school	1428 (36.9)	1820 (34.8)	
Upper college	1715 (46.2)	2023 (37.4)	
**Nutrient Intake (mean ± SD)**			
Food intake	1834.9 ± 18.9	1391.6 ± 12.2	<0.0001
Energy	2401.2 ± 20.3	1681.8 ± 11.9	<0.0001
Moisture	1105.5 ± 13.2	916.3 ± 9.8	<0.0001
Protein	87.3 ± 0.9	60.2 ± 0.5	<0.0001
Fat	55.5 ± 0.8	38.4 ± 0.5	<0.0001
Saturated fatty acids	18.0 ± 0.3	12.4 ± 0.2	<0.0001
Monounsaturated fatty acids	18.1 ± 0.3	12.2 ± 0.2	<0.0001
Omega-3 fatty acid	2.1 ± 0	1.6 ± 0	<0.0001
Omega-6 fatty acid	11.9 ± 0.2	8.5 ± 0.1	<0.0001
Cholesterol	299.1 ± 5.1	211.7 ± 3.2	<0.0001
Carbohydrates	337.3 ± 2.6	263.0 ± 1.8	<0.0001
Dietary fiber	27.0 ± 0.3	23.1 ± 0.2	<0.0001
Sugar	67.1 ± 1	57.8 ± 0.7	<0.0001
Calcium	582.3 ± 7.1	460.0 ± 5.2	<0.0001
Phosphorus	1251.1 ± 10.9	922.8 ± 6.9	<0.0001
Iron	13.9 ± 0.2	10.6 ± 0.1	<0.0001
Sodium	4143.1 ± 44.7	2816.7 ± 29.4	<0.0001
Potassium	3156.4 ± 29.0	2534.0 ± 20.4	<0.0001
Vitamin A	675.7 ± 13.5	563.7 ± 15.8	<0.0001
Carotene	2960.5 ± 53.5	2586.0 ± 90.5	<0.0001
Vitamin B1	1.6 ± 0.0	1.1 ± 0.0	<0.0001
Vitamin B2	1.9 ± 0.0	1.4 ± 0.0	<0.0001
Vitamin B3	16.0 ± 0.2	11.5 ± 0.1	<0.0001
Vitamin C	64.8 ± 1.6	59.6 ± 1.2	0.0010
**Dietary Habits (*N* (%))**			
Frequency of breakfast			0.1658
5–7/week	2796 (58.9)	3883 (60.6)	
3–4/week	433 (12.4)	639 (11.8)	
1–2/week	414 (12.2)	645 (12.6)	
<1/week	555 (16.4)	741 (15)	
Frequency of lunch			<0.0001
5–7/week	3862 (91.5)	5207 (87.6)	
3–4/week	188 (5.1)	462 (8.4)	
1–2/week	73 (1.8)	130 (2.2)	
<1/week	75 (1.6)	109 (1.8)	
Frequency of dinner			<0.0001
5–7/week	3962 (92.9)	5216 (86.9)	
3–4/week	186 (5.7)	537 (10.3)	
1–2/week	40 (1.2)	126 (2.3)	
<1/week	10 (0.2)	29 (0.5)	
Frequency of eating out			<0.0001
2>/day	492 (14.0)	188 (3.8)	
1/day	1004 (27.5)	710 (13.8)	
5–6/week	742 (19.8)	803 (14.4)	
3–4/week	441 (10.8)	730 (13.5)	
1–2/week	685 (14.5)	1630 (27.2)	
1/month	589 (9.8)	1275 (19.5)	
<1/month	245 (3.7)	572 (7.8)	

**Table 2 nutrients-13-01360-t002:** The results of univariate simple and multiple linear regression for nutrient intake and dietary habits as they relate to PHQ-9 scores of the study population.

	Simple		Multiple	
Variables	Standardized Coef. (95% CI)	*p*-Value	Standardized Coef. (95% CI)	*p*-Value
Energy	−2.5 × 10^−4^(−3.3 × 10^−4^–−1.7 × 10^−4^)	<0.0001	2.6 × 10^−4^(−1.0 × 10^−4^–6.2 × 10^−4^)	0.1592
Moisture	−3.8 × 10^−4^(−5.0 × 10^−4^–−2.5 × 10^−4^)	<0.0001	−2.5 × 10^−5^(−3.8 × 10^−4^–3.3 × 10^−4^)	0.8893
Protein	−5.9 × 10^−3^(−7.8 × 10^−3^–−4.0 × 10^−3^)	<0.0001	8.2 × 10^−4^(−6.2 × 10^−3^–7.8 × 10^−3^)	0.8186
Fat	−4.1 × 10^−3^(−5.9 × 10^−3^–−2.3 × 10^−3^)	<0.0001	0.014(−0.026–0.054)	0.4843
Saturated fatty acids	−0.01(−0.016–−0.005)	<0.0001	−0.007(−0.054–0.04)	0.7617
Monounsaturated fatty acids	−0.011(−0.016–−0.007)	<0.0001	−0.029(−0.079–0.021)	0.2499
Omega-3 fatty acid	−0.08(−0.125–−0.036)	0.0004	−4.1 × 10^−4^(−6.8 × 10^−2^–6.8 × 10^−2^)	0.9905
Omega-6 fatty acid	−0.017(−0.025–−0.009)	<0.0001	−0.01(−0.054–0.034)	0.6502
Cholesterol	−6.8 × 10^−4^(−1.0 × 10^−3^–−3.5 × 10^−4^)	<0.0001	−6.3 × 10^−4^(−1.2 × 10^−3^–−6.6 × 10^−5^)	**0.0287**
Carbohydrates	−1.8 × 10^−3^(−2.4 × 10^−3^–−1.2 × 10^−3^)	<0.0001	4.4 × 10^−4^(−1.3 × 10^−3^–2.1 × 10^−3^)	0.6118
Dietary fiber	−0.024(−0.029–−0.019)	<0.0001	−1.2 × 10^−2^(−2.3 × 10^−2^–−7.4 × 10^−4^)	**0.0365**
Sugar	−2.8 × 10^−3^(−4.8 × 10^−3^–−9.1 × 10^−4^)	0.004	−8.0 × 10^−4^(−3.9 × 10^−3^–2.3 × 10^−3^)	0.6132
Calcium	−7.1 × 10^−4^(−9.1 × 10^−4^–−5.0 × 10^−4^)	<0.0001	2.3 × 10^−5^(−3.0 × 10^−4^–3.4 × 10^−4^)	0.8863
Phosphorus	−5.6 × 10^−4^(−7.1 × 10^−4^–−4.0 × 10^−4^)	<0.0001	−2.3 × 10^−4^(−7.9 × 10^−4^–3.3 × 10^−4^)	0.4177
Iron	−4.3 × 10^−3^(−1.0 × 10^−2^–1.5 × 10^−3^)	0.1472	0.011(−0.005–0.028)	0.1761
Sodium	−1.0 × 10^−4^(−1.4 × 10^−4^–−7.3 × 10^−5^)	<0.0001	−7.1 × 10^−5^(−1.2 × 10^−4^–−2.6 × 10^−5^)	**0.0023**
Potassium	−2.0 × 10^−4^(−2.5 × 10^−4^–−1.6 × 10^−4^)	<0.0001	−4.9 × 10^−6^(−1.5 × 10^−4^–1.4 × 10^−4^)	0.9458
Vitamin A	−1.3 × 10^−4^(−2.0 × 10^−4^–−6.1 × 10^−5^)	0.0003	−1.2 × 10^−4^(−3.3 × 10^−4^–9.2 × 10^−5^)	0.2639
Carotene	−2.3 × 10^−5^(−3.8 × 10^−5^–−7.4 × 10^−6^)	0.0037	1.1 × 10^−6^(−3.9 × 10^−5^–4.1 × 10^−5^)	0.9563
Vitamin B1	−0.216(−0.292–−0.14)	<0.0001	−0.085(−0.232–0.063)	0.2585
Vitamin B2	−0.256(−0.333–−0.178)	<0.0001	0.19(−0.033–0.414)	0.0946
Vitamin B3	−0.022(−0.031–−0.012)	<0.0001	3.1 × 10^−3^(−1.6 × 10^−2^–2.2 × 10^−2^)	0.7468
Vitamin C	−1.0 × 10^−3^(−1.7 × 10^−3^–−3.7 × 10^−4^)	0.0023	−2.7 × 10^−4^(−1.4 × 10^−3^–8.6 × 10^−4^)	0.635
Frequency of breakfast				
5–7/week	0.0(0.0–0.0)		0.0(0.0–0.0)	
3–4/week	0.151(−0.045–0.347)	0.1319	0.025(−0.201–0.25)	0.8312
1–2/week	0.601(0.385–0.816)	<0.0001	0.488(0.247–0.729)	**<0.0001**
0/week	0.924(0.704–1.145)	<0.0001	0.781(0.536–1.026)	**<0.0001**
Frequency of lunch				
5–7/week	0.0(0.0–0.0)		0.0(0.0–0.0)	
3–4/week	1.208(0.889–1.526)	<0.0001	0.948(0.56–1.336)	**<0.0001**
1–2/week	2.125(1.434–2.816)	<0.0001	1.602(0.772–2.431)	**0.0002**
0/week	1.468(0.733–2.202)	<0.0001	0.733(−0.131–1.597)	0.0963
Frequency of dinner				
5–7/week	0.0(0.0–0.0)		0.0(0.0–0.0)	
3–4/week	0.698(0.418–0.977)	<0.0001	0.244(−0.061–0.55)	0.1164
1–2/week	1.227(0.587–1.867)	0.0002	0.831(0.089–1.574)	**0.0283**
0/week	1.447(−0.029–2.923)	0.0547	0.904(−0.444–2.252)	0.1883
Frequency of eating out				
2>/day	−0.343(−0.62–−0.066)	0.0155	−3.5 × 10^−3^(−0.34–0.33)	0.9838
1/day	−0.345(−0.548–−0.143)	0.0009	0.047(−0.204–0.299)	0.7122
5–6/week	−0.315(−0.539–−0.091)	0.006	−0.159(−0.425–0.107)	0.2415
3–4/week	0.117(−0.128–0.361)	0.3481	0.048(−0.225–0.32)	0.7314
1–2/week	0.0(0.0–0.0)		0.0(0.0–0.0)	
1–3/month	0.294(0.034–0.553)	0.0265	0.203(−0.1–0.506)	0.1886
1</month	1.226(0.838–1.614)	<0.0001	1.073(0.613–1.533)	**<0.0001**

Note: We provide the regression coefficient, *p*-value, and 95% confidence interval. We used total food intake, gender, income level, education level, and marital status as adjusted variables in the multiple regression. Bold values denote statistical significance with *p*-values less than 0.05.

**Table 3 nutrients-13-01360-t003:** Multiple linear regression results for the men’s groups.

	Men			
Variables	19–34	35–49	50–64	≥65
	Standardized Coef. (95% CI)	Standardized Coef. (95% CI)	Standardized Coef. (95% CI)	Standardized Coef. (95% CI)
Energy	2.6 × 10^−^^4^ (−5.0 × 10^−^^4^–1.0 × 10^−^^3^)	5.5 × 10^−^^5^ (−5.6 × 10^−^^4^–6.6 × 10^−^^4^)	**1.0** **× 10** **^−^** **^3^** **(4.4** **× 10** **^−^** **^4^** **–1.6** **× 10** **^−^** **^3^** **) ****	3.0 × 10^−^^4^ (−1.1 × 10^−^^3^–1.7 × 10^−^^3^)
Moisture	4.6 × 10^−^^4^ (−3.3 × 10^−^^4^–1.3 × 10^−^^3^)	−4.7 × 10^−^^4^ (−1.1 × 10^−^^3^–1.2 × 10^−^^4^)	**9.3** **× 10** **^−^** **^4^** **(2.8** **× 10** **^−^** **^4^** **–1.6** **× 10** **^−^** **^3^** **) ****	−6.4 × 10^−^^5^ (−1.3 × 10^−^^3^–1.2 × 10^−^^3^)
Protein	1.3 × 10^−^^3^ (−1.4 × 10^−^^2^–1.6 × 10^−^^2^)	−5.1 × 10^−^^5^ (−1.3 × 10^−^^2^–1.3 × 10^−^^2^)	−2.2 × 10^−^^3^ (−1.8 × 10^−^^2^–1.4 × 10^−^^2^)	0.014 (−0.01–0.038)
Fat	0.095 (−0.017–0.207)	−0.036 (−0.098–0.025)	0.017 (−0.083–0.117)	0.044 (−0.066–0.154)
Saturated fatty acids	−0.123 (−0.26–0.015)	0.048 (−0.03–0.127)	−0.006 (−0.137–0.126)	−0.031 (−0.173–0.11)
Monounsaturated fatty acids	−0.093 (−0.223–0.037)	0.031 (−0.048–0.11)	−0.051 (−0.169–0.066)	−0.072 (−0.211–0.068)
Omega-3 fatty acid	−0.191 (−0.425–0.043)	0.136 (−0.042–0.315)	−0.059 (−0.211–0.094)	0.057 (−0.147–0.262)
Omega-6 fatty acid	−0.085 (−0.207–0.037)	0.029 (−0.038–0.096)	−0.044 (−0.158–0.07)	−0.018 (−0.152–0.115)
Cholesterol	−7.2 × 10^−^^4^ (−2.3 × 10^−^^3^–8.8 × 10^−^^4^)	−3.4 × 10^−^^4^ (−1.2 × 10^−^^3^–4.7 × 10^−^^4^)	−1.1 × 10^−^^3^ (−2.6 × 10^−^^3^–3.6 × 10^−^^4^)	3.5 × 10^−^^4^ (−1.4 × 10^−^^3^–2.1 × 10^−^^3^)
Carbohydrates	3.0 × 10^−^^3^ (−1.1 × 10^−^^3^–7.0 × 10^−^^3^)	−6.9 × 10^−^^4^ (−3.8 × 10^−^^3^–2.4 × 10^−^^3^)	**−** **3.1** **× 10** **^−^** **^3^** **(** **−** **5.9** **× 10** **^−^** **^3^** **–** **−** **3.7** **× 10** **^−^** **^4^** **) ***	−5.8 × 10^−^^4^ (−6.2 × 10^−^^3^–5.0 × 10^−^^3^)
Dietary fiber	−0.009 (−0.04–0.023)	−0.005 (−0.032–0.021)	−1.6 × 10^−^^3^ (−2.7 × 10^−^^2^–2.3 × 10^−^^2^)	−0.006 (−0.032–0.02)
Sugar	−5.6 × 10^−^^3^ (−1.3 × 10^−^^2^–1.7 × 10^−^^3^)	4.0 × 10^−^^3^ (−1.8 × 10^−^^3^–9.8 × 10^−^^3^)	4.3 × 10^−^^3^ (−2.1 × 10^−^^3^–1.1 × 10^−^^2^)	1.2 × 10^−^^3^ (−7.8 × 10^−^^3^–1.0 × 10^−^^2^)
Calcium	3.9 × 10^−^^4^ (−8.0 × 10^−^^4^–1.6 × 10^−^^3^)	−1.6 × 10^−^^4^ (−1.0 × 10^−^^3^–7.2 × 10^−^^4^)	5.6 × 10^−^^4^ (−3.5 × 10^−^^4^–1.5 × 10^−^^3^)	7.1 × 10^−^^4^ (−2.1 × 10^−^^4^–1.6 × 10^−^^3^)
Phosphorus	−2.9 × 10^−^^4^ (−1.5 × 10^−^^3^–9.4 × 10^−^^4^)	2.7 × 10^−^^5^ (−1.2 × 10^−^^3^–1.2 × 10^−^^3^)	5.2 × 10^−^^4^ (−1.0 × 10^−^^3^–2.0 × 10^−^^3^)	−1.3 × 10^−^^3^ (−3.2 × 10^−^^3^–6.4 × 10^−^^4^)
Sodium	−5.6 × 10^−^^5^ (−2.1 × 10^−^^4^–9.7 × 10^−^^5^)	**−** **1.3** **× 10** **^−^** **^4^** **(** **−** **2.2** **×10** **^−^** **^4^** **–** **−** **3.8** **×10** **^−^** **^5^** **) ****	3.7 × 10^−^^5^ (−8.0 × 10^−^^5^–1.5 × 10^−^^4^)	−9.7 × 10^−^^5^ (−2.0 × 10^−^^4^–7.7 × 10^−^^6^)
Potassium	−1.2 × 10^−^^4^ (−6.4 × 10^−^^4^–4.0 × 10^−^^4^)	−1.3 × 10^−^^4^ (−4.5 × 10^−^^4^–1.8 × 10^−^^4^)	−1.1 × 10^−^^4^ (−3.5 × 10^−^^4^–1.2 × 10^−^^4^)	5.6 × 10^−^^5^ (−2.9 × 10^−^^4^–4.0 × 10^−^^4^)
Vitamin A	−3.8 × 10^−^^5^ (−6.7 × 10^−^^4^–5.9 × 10^−^^4^)	−2.9 × 10^−^^4^ (−7.9 × 10^−^^4^–2.1 × 10^−^^4^)	−5.1 × 10^−^^5^ (−4.1 × 10^−^^4^–3.0 × 10^−^^4^)	**−** **1.6** **× 10** **^−^** **^3^** **(** **−** **2.9** **× 10** **^−^** **^3^** **–** **−** **2.3** **× 10** **^−^** **^4^** **) ***
Carotene	−3.3 × 10^−^^5^ (−2.1 × 10^−^^4^–1.4 × 10^−^^4^)	3.2 × 10^−^^5^ (−6.9 × 10^−^^5^–1.3 × 10^−^^4^)	−2.4 × 10^−^^5^ (−1.1 × 10^−^^4^–5.7 × 10^−^^5^)	**3.1** **× 10** **^−^** **^4^** **(7.1** **× 10** **^−^** **^5^** **–5.5** **× 10** **^−^** **^4^** **) ***
Vitamin B1	−0.24 (−0.589–0.11)	0.106 (−0.209–0.42)	−0.095 (−0.487–0.296)	−0.044 (−0.709–0.62)
Vitamin B2	0.447 (−0.234–1.127)	−0.049 (−0.345–0.248)	0.099 (−0.368–0.566)	0.02 (−0.517–0.558)
Vitamin B3	1.9 × 10^−^^3^ (−3.6 × 10^−^^2^–4.0 × 10^−^^2^)	0.01 (−0.026–0.046)	0.013 (−0.037–0.062)	−4.9 × 10^−^^2^ (−0.1–3.6 × 10^−^^3^)
Vitamin C	−3.3 × 10^−^^4^ (−3.4 × 10^−^^3^–2.7 × 10^−^^3^)	−7.0 × 10^−^^4^ (−2.7 × 10^−^^3^–1.3 × 10^−^^3^)	−8.4 × 10^−^^4^ (−3.5 × 10^−^^3^–1.8 × 10^−^^3^)	2.9 × 10^−^^3^ (−2.3 × 10^−^^4^–6.0 × 10^−^^3^)
Frequency of breakfast				
5–7/week	0.0 (0.0–0.0)	0.0 (0.0–0.0)	0.0 (0.0–0.0)	0.0 (0.0–0.0)
1–2/week	0.151 (−0.641–0.943)	0.467 (−0.022–0.957)	0.403 (−0.418–1.225)	1.798 (−1.779–5.376)
0/week	0.268 (−0.441–0.978)	0.592 (−0.049–1.232)	0.22 (−0.471–0.911)	0.645 (−0.534–1.824)
Frequency of lunch				
5–7/week	0.0 (0.0–0.0)	0.0 (0.0–0.0)	0.0 (0.0–0.0)	0.0 (0.0–0.0)
3–4/week	0.101 (−0.691–0.892)	**1.236** **(0.212–2.26) ***	**1.171** **(0.139–2.203) ***	1.575 (−0.732–3.881)
1–2/week	0.543 (−1.219–2.304)	1.635 (0.127–3.143)	2.153 (−0.167–4.472)	**−** **0.931** **(** **−** **1.856–** **−** **0.005** **)** *****
0/week	−0.752 (−2.283–0.779)	0.951 (−1.67–3.572)	1.601 (−0.902–4.105)	−0.51 (−1.516–0.495)
Frequency of dinner				
5–7/week	0.0 (0.0–0.0)	0.0 (0.0–0.0)	0.0 (0.0–0.0)	0.0 (0.0–0.0)
3–4/week	−0.185 (−0.815–0.444)	−0.587 (−1.5–0.326)	**1.124** **(0.257–1.991) ***	0.079 (−1.258–1.416)
1–2/week	1.516 (−0.739–3.772)	0.476 (−1.745–2.696)	**1.359** **(** **−** **1.68–4.397) ***	2.705 (−2.526–7.936)
Frequency of eating out				
2>/day	−0.296 (−1.391–0.799)	0.076 (−0.747–0.899)	−0.02 (−0.621–0.581)	1.123 (−0.28–2.526)
1/day	−0.526 (−1.572–0.521)	0.334 (−0.432–1.1)	0.16 (−0.37–0.69)	0.061 (−0.626–0.748)
5–6/week	−0.395 (−1.411–0.622)	0.096 (−0.727–0.919)	−0.223 (−0.758–0.312)	0.189 (−0.473–0.852)
1–2/week	0.0 (0.0–0.0)	0.0 (0.0–0.0)	0.0 (0.0–0.0)	0.0 (0.0–0.0)
1–3/month	0.583 (−1.209–2.374)	0.076 (−0.747–0.899)	0.539 (−0.309–1.388)	0.434 (−0.031–0.898)
1</month	0.696 (−2.588–3.981)	0.334 (−0.432–1.1)	1.481 (−0.069–3.03)	**1.457** **(0.617–2.298) ****

Note: We provide nutritional intake and eating habits associated with depression according to the age of the men. We used total food intake, income level, education level, and marital status as adjusted variables in the multiple regression. Bold values indicate statistical significance with *p*-values less than 0.05. * *p*-value < 0.05, ** *p*-value < 0.01.

**Table 4 nutrients-13-01360-t004:** Multiple linear regression results for the women’s groups.

	Women			
Variables	19–34	35–49	50–64	≥65
	Standardized coef. (95% CI)	Standardized coef. (95% CI)	Standardized coef. (95% CI)	Standardized coef. (95% CI)
Energy	−6.6 × 10^−^^4^ (−2.3 × 10^−^^3^–9.9 × 10^−^^4^)	−2.6 × 10^−^^6^ (−1.3 × 10^−^^3^–1.3 × 10^−^^3^)	−1.7 × 10^−^^4^ (−1.8 × 10^−^^3^–1.5 × 10^−^^3^)	−2.9 × 10^−^^3^ (−8.4 × 10^−^^3^–2.6 × 10^−^^3^)
Moisture	−1.1 × 10^−^^3^ (−2.2 × 10^−^^3^–1.1 × 10^−^^4^)	5.7 × 10^−^^6^ (−7.9 × 10^−^^4^–8.0 × 10^−^^4^)	−8.2 × 10^−^^4^ (−2.4 × 10^−^^3^–7.6 × 10^−^^4^)	8.0 × 10^−^^4^ (−1.6 × 10^−^^3^–3.2 × 10^−^^3^)
Protein	−0.006 (−0.028–0.015)	−1.9 × 10^−^^3^ (−1.7 × 10^−^^2^–1.3 × 10^−^^2^)	2.1 × 10^−^^3^ (−1.8 × 10^−^^2^–2.3 × 10^−^^2^)	0.031 (−0.01–0.072)
Fat	0.081 (−0.066–0.228)	0.055 (−0.033–0.143)	−0.106 (−0.231–0.02)	**−** **0.189** **(** **−** **0.369–** **−** **0.009) ***
Saturated fatty acids	−0.066 (−0.232–0.101)	−0.069 (−0.174–0.036)	0.137 (−0.02–0.295)	**0.291** **(0.073–0.508) ****
Monounsaturated fatty acids	−0.112 (−0.276–0.053)	−0.039 (−0.144–0.066)	0.093 (−0.048–0.234)	0.146 (−0.067–0.358)
Omega-3 fatty acid	−0.037 (−0.335–0.262)	−0.025 (−0.151–0.102)	0.093 (−0.061–0.248)	**0.358** **(0.071–0.645) ***
Omega-6 fatty acid	−0.012 (−0.182–0.158)	−0.082 (−0.177–0.012)	0.113 (−0.013–0.239)	**0.241** **(0.048–0.435) ***
Cholesterol	−5.0 × 10^−^^4^ (−2.2 × 10^−^^3^–1.2 × 10^−^^3^)	−6.4 × 10^−^^4^ (−1.9 × 10^−^^3^–6.3 × 10^−^^4^)	−3.7 × 10^−^^4^ (−1.9 × 10^−^^3^–1.2 × 10^−^^3^)	−7.8 × 10^−^^4^ (−3.5 × 10^−^^3^–1.9 × 10^−^^3^)
Carbohydrates	3.7 × 10^−^^3^ (−4.3 × 10^−^^3^–1.2 × 10^−^^2^)	1.9 × 10^−^^3^ (−3.4 × 10^−^^3^–7.2 × 10^−^^3^)	4.3 × 10^−^^3^ (−2.5 × 10^−^^3^–1.1 × 10^−^^2^)	0.01 (−0.012–0.032)
Dietary fiber	−0.031 (−0.067–0.005)	**−** **0.031** **(** **−** **0.055–** **−** **0.006) ***	−0.01 (−0.034–0.014)	−0.016 (−0.052–0.021)
Sugar	**−** **1.5** **× 10** **^−^** **^2^** **(** **−** **2.5** **× 10** **^−^** **^2^** **–** **−** **4.9** **× 10** **^−^** **^3^** **) ****	5.8 × 10^−^^3^ (−2.4 × 10^−^^3^–1.4 × 10^−^^2^)	4.1 × 10^−^^4^ (−8.5 × 10^−^^3^–9.3 × 10^−^^3^)	4.2 × 10^−^^3^ (−1.1 × 10^−^^2^–1.9 × 10^−^^2^)
Calcium	−1.7 × 10^−^^4^ (−1.3 × 10^−^^3^–1.0 × 10^−^^3^)	−2.8 × 10^−^^4^ (−8.5 × 10^−^^4^–2.9 × 10^−^^4^)	−3.6 × 10^−^^4^ (−1.2 × 10^−^^3^–4.4 × 10^−^^4^)	−4.9 × 10^−^^4^ (−2.2 × 10^−^^3^–1.2 × 10^−^^3^)
Phosphorus	−5.5 × 10^−^^4^ (−2.3 × 10^−^^3^–1.2 × 10^−^^3^)	7.2 × 10^−^^4^ (−6.8 × 10^−^^4^–2.1 × 10^−^^3^)	−5.0 × 10^−^^4^ (−2.0 × 10^−^^3^–1.0 × 10^−^^3^)	−1.1 × 10^−^^3^ (−3.8 × 10^−^^3^–1.6 × 10^−^^3^)
Sodium	−1.1 × 10^−^^4^ (−2.9 × 10^−^^4^–6.7 × 10^−^^5^)	−5.7 × 10^−^^5^ (−2.0 × 10^−^^4^–8.3 × 10^−^^5^)	−8.0 × 10^−^^5^ (−2.0 × 10^−^^4^–4.0 × 10^−^^5^)	−2.2 × 10^−^^5^ (−2.3 × 10^−^^4^–1.8 × 10^−^^4^)
Potassium	1.1 × 10^−^^4^ (−3.3 × 10^−^^4^–5.6 × 10^−^^4^)	1.0 × 10^−^^4^ (−3.3 × 10^−^^4^–5.2 × 10^−^^4^)	4.8 × 10^−^^5^ (−3.1 × 10^−^^4^–4.1 × 10^−^^4^)	−9.6 × 10^−^^6^ (−4.3 × 10^−^^4^–4.1 × 10^−^^4^)
Vitamin A	−4.4 × 10^−^^4^ (−1.3 × 10^−^^3^–3.9 × 10^−^^4^)	7.0 × 10^−^^5^ (−4.0 × 10^−^^4^–5.4 × 10^−^^4^)	−8.4 × 10^−^^4^ (−1.8 × 10^−^^3^–1.4 × 10^−^^4^)	4.0 × 10^−^^5^ (−2.0 × 10^−^^3^–2.1 × 10^−^^3^)
Carotene	4.0 × 10^−^^5^ (−9.6 × 10^−^^5^–1.8 × 10^−^^4^)	−1.0 × 10^−^^4^ (−2.2 × 10^−^^4^–1.9 × 10^−^^5^)	1.7 × 10^−^^4^ (−1.5 × 10^−^^5^–3.5 × 10^−^^4^)	2.8 × 10^−^^6^ (−3.4 × 10^−^^4^–3.4 × 10^−^^4^)
Vitamin B1	0.258 (−0.217–0.732)	−0.123 (−0.434–0.187)	−0.062 (−0.484–0.36)	−0.534 (−1.282–0.214)
Vitamin B2	0.19 (−0.405–0.785)	0.163 (−0.254–0.579)	0.135 (−0.428–0.698)	0.591 (−0.231–1.413)
Vitamin B3	0.008 (−0.066–0.082)	−0.03 (−0.074–0.015)	0.038 (−0.021–0.096)	−0.023 (−0.143–0.098)
Vitamin C	1.2 × 10^−^^3^ (−3.0 × 10^−^^3^–5.3 × 10^−^^3^)	−1.4 × 10^−^^4^ (−2.7 × 10^−^^3^–2.4 × 10^−^^3^)	6.0 × 10^−^^4^ (−3.0 × 10^−^^3^–4.2 × 10^−^^3^)	6.2 × 10^−^^4^ (−5.6 × 10^−^^3^–6.9 × 10^−^^3^)
Frequency of breakfast				
5–7/week	0.0 (0.0–0.0)	0.0 (0.0–0.0)	0.0 (0.0–0.0)	0.0 (0.0–0.0)
1–2/week	**1.078** **(0.438–1.719) ****	0.291 (−0.193–0.775)	0.188 (−0.479–0.855)	−0.18 (−1.653–1.292)
0/week	**1.221** **(0.607–1.835) ****	**0.795** **(0.228–1.362) ****	**1.03** **(0.315–1.745) ****	1.319 (−0.608–3.245)
Frequency of lunch				
5–7/week	0.0 (0.0–0.0)	0.0 (0.0–0.0)	0.0 (0.0–0.0)	0.0 (0.0–0.0)
3–4/week	0.786 (−0.076–1.648)	0.615 (−0.131–1.362)	**1.235** **(0.124–2.346) ***	1.062 (−0.243–2.367)
1–2/week	**3.233** **(0.631–5.836) ***	0.366 (−0.521–1.254)	**1.852** **(0.305–3.399) ***	2.19 (−1.048–5.428)
0/week	3.373 (−0.249–6.994)	2.115 (−0.837–5.067)	4.6 × 10^−^^3^ (−1.4–1.5)	−0.96 (−2.585–0.666)
Frequency of dinner				
5–7/week	0.0 (0.0–0.0)	0.0 (0.0–0.0)	0.0 (0.0–0.0)	0.0 (0.0–0.0)
3–4/week	0.233 (−0.51–0.976)	0.168 (−0.465–0.802)	0.673 (−0.147–1.492)	0.265 (−1.147–1.678)
1–2/week	−0.035 (−1.371–1.301)	0.749 (−0.45–1.948)	0.565 (−0.666–1.795)	2.738 (−0.55–6.027)
Frequency of eating out				
2>/day	0.685 (−0.367–1.737)	0.052 (−1.047–1.15)	−0.258 (−1.166–0.651)	**−** **1.609** **(** **−** **2.712–** **−** **0.506) ****
1/day	0.591 (−0.119–1.301)	**−** **0.649** **(** **−** **1.176–** **−** **0.123) ***	−0.369 (−1.018–0.279)	0.301 (−1.038–1.641)
5–6/week	0.235 (−0.518–0.989)	**−** **0.591** **(** **−** **1.106–** **−** **0.075) ***	−0.241 (−0.927–0.444)	0.182 (−0.866–1.229)
1–2/week	0.0 (0.0–0.0)	0.0 (0.0–0.0)	0.0 (0.0–0.0)	0.0 (0.0–0.0)
1–3/month	0.528 (−0.503–1.56)	−0.479 (−1.022–0.065)	0.372 (−0.197–0.94)	−0.061 (−0.712–0.591)
1</month	**3.821** **(0.355–7.287) ***	1.424 (−1.238–4.086)	0.75 (−0.272–1.773)	0.656 (−0.107–1.42)

Note: We provide nutritional intake and eating habits associated with depression according to the age of the women. We used total food intake, income level, education level, and marital status as adjusted variables in the multiple regression. Bold values indicate statistical significance with *p*-values less than 0.05. * *p*-value < 0.05, ** *p*-value < 0.01.

**Table 5 nutrients-13-01360-t005:** The results of factor rotation.

Variables	Factor1	Factor2	Factor3
Fat	**0.95**	0.17	0.07
Monounsaturated fatty acids	**0.92**	0.1	0.07
Saturated fatty acids	**0.89**	0.1	0.02
Protein	**0.75**	0.5	0.12
Omega-6 fatty acid	**0.74**	0.28	0.09
Cholesterol	**0.72**	0.16	0.19
Energy	**0.69**	0.62	0.02
Vitamin B2	**0.66**	0.48	0.25
Phosphorus	**0.65**	0.64	0.17
Vitamin B3	**0.63**	0.52	0.17
Vitamin B1	**0.54**	0.51	0.17
Omega-3 fatty acid	**0.41**	0.3	0.27
Dietary fiber	0.05	**0.87**	0.15
Carbohydrates	0.27	**0.82**	−0.05
Potassium	0.33	**0.82**	0.25
Moisture	0.24	**0.72**	0.18
Sugar	0.24	**0.64**	−0.03
Iron	0.36	**0.63**	0.22
Sodium	0.45	**0.55**	0.11
Calcium	0.34	**0.53**	0.26
Vitamin C	−0.03	**0.5**	0.13
Vitamin A	0.2	0.3	**0.89**
Carotene	0	0.34	**0.86**

Bold values indicate factor loading of greater than 0.4.

**Table 6 nutrients-13-01360-t006:** Multiple linear regression results using the results of factor analysis for entire, men’s, and women’s groups.

Groups	Variables	Coef. (95% CI)	*p*-Value
Entire group	Factor1	0.012(−0.062–0.086)	0.7484
	Factor2	−0.11(−0.187–−0.033)	**0.0052**
	Factor3	−0.096(−0.152–−0.041)	**0.0006**
Men’s group	Factor1	−0.004(−0.094–0.086)	0.928
	Factor2	−0.066(−0.161–0.028)	0.1689
	Factor3	−0.112(−0.201–−0.024)	**0.0132**
Women’s group	Factor1	0.055(−0.083–0.193)	0.4333
	Factor2	−0.169(−0.295–−0.044)	**0.0084**
	Factor3	−0.092(−0.156–−0.028)	**0.0048**

Note: We used income level, education level, and marital status as adjusted variables in multiple regression. Bold values denote statistical significance with *p*-values less than 0.05.

## Data Availability

The data presented in this study are available on request from the corresponding author.

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
