# Peer review of "Analysis of the Effects of Nutrient Intake and Dietary Habits on Depression in Korean Adults"

_nutrients, 2021, doi:10.3390/nu13041360_

Round 1

Reviewer 1 Report

  1. What is the reference group for variables “Frequency of breakfast / lunch /dinner/ eating out”? As the majority of subjects fall into the highest frequency level for the meals, use of any other group would not be appropriate (sample size too small as a reference group).
  2. Multiple testing is a big concern due to a long list of nutrients included in the analysis.
  3.  Are the nutrients intake adjusted for total caloric intake?
  4.  The beta coefficients for the nutrients are extremely small. Are these effect size meaningful in terms of prevention?
  5.  Does it make sense that total unsaturated fatty acids has an opposite relationship with depression than its two components omega-3 and omega-6? (Figure 3)
  6.  Are the models (table 2 and 3) adjusted for other potential confounders such as income, education, etc?
  7.  Figure 3 information is redundant as the results can be found in the tables.

Reviewer 2 Report

General comments: This manuscript assessed the association between nutrient intake and prevalent depression. Although it is hard which one causes which (reverse causation) in this cross-sectional analysis, it does not detract the merit of the study and I would like to congratulate the authors for their very good work. I have the following few point for authors to consider/include in thee manuscript. Major comment: Method and results: I recommend the authors to identify nutrient patterns using factor analysis. This approach can handle synergistic or interaction effect of nutrients on depression. In addition it would be very interesting if the author explore the effect of nutrients and nutrient patterns on individual component of depression. The author also use the depression score as a continuous variable. Did they transform the score? did they check model assumptions in linear regression. In addition, to provide meaningful estimates, it is always clinically important to categorize depression score into depressed/not depress and run a logistic/Poisson regression. I also found the result hard to follow - that is why factor analysis will not only avoid this but also considers the interaction effect of nutrients. Minor comments: Figure 2 - what does average mean (mean or median). This should be median as the distribution is not quite symmetric. Discussion: it is important to discuss the mechanism of action of nutrients on brain/mood/depression

Round 2

Reviewer 1 Report

The manuscript has been improved with previous critiques. No further comments.

Reviewer 2 Report

The authors have addressed all my concerns.